# Polarization-state-resolved high-harmonic spectroscopy of solids

N. Klemke [1,2], N. Tancogne-Dejean [1,3], G.M. Rossi [1,2], Y. Yang [1,2], F. Scheiba [1,2], R.E. Mainz[1,2], G. Di Sciacca [1], A. Rubio [1,2,3,4,5], F.X. Kärtner[1,2,4] & O.D. Mücke [1,4]

Attosecond metrology sensitive to sub-optical-cycle electronic and structural dynamics is opening up new avenues for ultrafast spectroscopy of condensed matter. Using intense lightwaves to precisely control the fast carrier dynamics in crystals holds great promise for next-generation petahertz electronics and devices. The carrier dynamics can produce high-order harmonics of the driving field extending up into the extreme-ultraviolet region. Here, we introduce polarization-state-resolved high-harmonic spectroscopy of solids, which provides deeper insights into both electronic and structural sub-cycle dynamics. Performing high-harmonic generation measurements from silicon and quartz, we demonstrate that the polarization states of the harmonics are not only determined by crystal symmetries, but can be dynamically controlled, as a consequence of the intertwined interband and intraband electronic dynamics. We exploit this symmetry-dynamics duality to efficiently generate coherent circularly polarized harmonics from elliptically polarized pulses. Our experimental results are supported by ab-initio simulations, providing evidence for the microscopic origin of the phenomenon.

[1] Center for Free-Electron Laser Science CFEL, Deutsches Elektronen-Synchrotron DESY, Notkestraße 85, 22607 Hamburg, Germany. [2] Physics Department, University of Hamburg, Luruper Chaussee 149, 22761 Hamburg, Germany. [3] Max Planck Institute for the Structure and Dynamics of Matter, Luruper Chaussee 149, 22761 Hamburg, Germany. [4] The Hamburg Centre for Ultrafast Imaging, Luruper Chaussee 149, 22761 Hamburg, Germany. [5] Center for Computational Quantum Physics (CCQ), The Flatiron Institute, 162 Fifth Avenue, New York, NY 10010, USA. These authors contributed equally: N. Klemke, N. Tancogne-Dejean. Correspondence and requests for materials should be addressed to N.T.-D. (email: nicolas.tancogne-dejean@mpsd.mpg.de) or to A.R. (email: angel.rubio@mpsd.mpg.de) or to O.D.M. (email: oliver.muecke@cfel.de)

The study of lightwave-driven electronic dynamics occurring on sub-optical-cycle time scales in condensed matter and nanosystems is a fascinating frontier of attosecond science originally developed in atoms and molecules[1]. Adapting attosecond metrology techniques[2] to observe and control the fastest electronic dynamics in the plethora of known solids and novel quantum materials[3] is very promising for studying correlated electronic dynamics (e.g., excitonic effects, screening) on atomic length and time scales, thereby potentially impacting future technologies such as emerging petahertz electronic signal processing[2,4] or strong-field optoelectronics[5,6].

The nonlinear process of high-order harmonic generation (HHG) in gases is one of the cornerstones of attosecond science and is well understood by the semiclassical three-step model[1]. In solids, nonperturbative HHG up to 25th harmonic order without irreversible damage was first reported in ref. [7]. This work triggered extensive research activities aimed at unraveling the microscopic interband and intraband dynamics underlying HHG from crystals (for a comprehensive overview, see ref. [8]), thereby extending attoscience techniques to solids. The prevailing strong-field dynamics were successfully identified in specific cases, even if a global picture has not yet emerged. Other works demonstrated isolated attosecond extreme ultraviolet (XUV) pulses emitted from thin $SiO_2$ films[9], or investigated HHG from two-dimensional (2D) materials such as graphene[10], 2D transition-metal dichalcogenides[10,11], and monolayer hexagonal boron nitride[12].

Elucidating the complex microscopic electronic dynamics producing HHG without making a priori severe assumptions poses a challenge for theory. Indeed, the theory must capture at the same time the transitions between discrete electronic bands, and the ultrafast motion of electrons within the bands; two mechanisms usually decoupled in the description of either optical properties or transport in semiconductors and insulators. An effective way to account for the full interacting many-body electronic dynamics and real crystal structure is using ab-initio time-dependent density functional theory (TDDFT) simulations[13–15]. Some of us recently used this theoretical framework to reveal how the microscopic mechanisms governing HHG in solids depend on the ellipticity of the driving field and the underlying band structure[15]. That work predicted that different harmonics react differently to the driver ellipticity, as they can either originate mainly from intraband contributions or from coupled interband and intraband dynamics[14].

The symmetry properties of the light-matter interaction Hamiltonian distinguishes HHG in crystals from atoms and molecules, with major ramifications for the selection rules of different harmonics and their polarization states. HHG from atoms driven by propeller-shaped bichromatic waveforms produces circular harmonics[16] (for convenience, we often use the terminology linear/circular/elliptical instead of linearly/circularly/elliptically polarized to refer to the polarization state). In molecules, both the point group and the driving field determine the symmetries of the coupled light-matter system. Consequently, depending on the molecular symmetries and the molecule's orientation with respect to the light polarization direction, elliptical high-order harmonics can be produced by linear[17] or elliptical driver pulses[18]. For bichromatic bicircular driver fields, circular harmonics with alternating helicities can be obtained, provided that the molecule's symmetries are compatible with that of the driving field[19]. In crystals, several recent works studied the high-harmonic response on driver pulse ellipticity $\varepsilon$, which can strongly differ qualitatively from the atomic and molecular cases. Whereas earlier work[20] looked exclusively at the harmonic yield, later research also investigated the polarization states and selection rules of the higher harmonics from various solids of different

crystal symmetries[15,21,22] and reported circular HHG from a single-color driver field[21,22], which is symmetry forbidden in atoms.

Here, we present a combination of HHG experiments and first-principles TDDFT simulations for silicon and quartz, demonstrating that a complete understanding of the harmonics' polarization states requires, beside knowledge of the crystal's symmetries, a microscopic understanding of the underlying complex, coupled interband and intraband dynamics[14,15]. Most importantly, we demonstrate strong-field control of the harmonics' polarization states. Our findings indicate that polarization-state-resolved high-harmonic spectroscopy of solids provides deeper insights into both electronic and structural dynamics as well as symmetries on sub-cycle time scales. This spectroscopy technique therefore might find important applications in future studies of novel quantum materials[3] such as strongly correlated materials[23,24], topological insulators[25], and magnetic materials[26]. Moreover, compact sources of bright circularly polarized harmonics in the XUV regime might advance our tools for the spectroscopy of chiral systems[18] and 2D materials with valley selectivity[11].

## Results

**High-harmonic generation experiments**. In our experiments, we irradiated free-standing, 2-μm-thin, (100)-cut silicon samples with 120-fs, 2.1-μm (0.59 eV) pulses with tunable ellipticity $\varepsilon$ and peak intensities up to 0.7 TW cm$^{-2}$ in vacuum (see Methods section and Supplementary Figure 1). At this intensity, the harmonics are generated nonperturbatively (see Supplementary Figure 3) up to harmonic order 19 (HH19) in the XUV regime for our experimental conditions, as shown by our TDDFT simulations (see Supplementary Figure 5). Only harmonics up to HH9 are detected by the spectrometer used in our experiments. We also irradiated 50-μm-thin, z-cut quartz with an estimated intensity of 40 TW cm$^{-2}$ in vacuum.

Figure 1 shows the measured high-harmonic response in Si of HH5, HH7, and HH9 as a function of driver ellipticity $\varepsilon$ and sample orientation $\theta$; panels a–c display normalized harmonic intensities, panels d–f harmonic ellipticities $|\varepsilon_{HH}| = \sqrt{I_{min}/I_{max}}$, where $I_{min}$ and $I_{max}$ correspond to the intensities at the minor and major axes of the polarization ellipse. In all panels, $\theta = 0°$, 45° refer to the major axis of the driving field ellipse along the directions [100] (ΓX) and [110] (ΓK) in real (reciprocal) space. The crystal symmetries are recovered in all maps shown in Fig. 1.

All harmonics respond in a distinctly different way to the driver pulse ellipticity $\varepsilon$, and the harmonic yields peak for different sample rotations. HH5 exhibits monotonically decreasing yield versus ellipticity profiles for all sample rotations, resembling the Gaussian-shaped profile in the atomic case[27]. We therefore call such profiles atomic-like. The distribution is symmetric around $\varepsilon = 0$ (see white dotted center-of-mass curve) for all sample rotations. The intensity distribution of HH7 (Fig. 1b) shows intriguing, non-atomic-like features with maximum yield at non-zero ellipticity for certain sample rotations, similar to experiments on MgO[20]. HH9 (Fig. 1c) exhibits the most pronounced deviations from a Gaussian-like ellipticity profile, with non-monotonic non-atomic-like profiles for wide ranges of sample rotation. Its yield is strongly asymmetric with respect to $\varepsilon = 0$ for all sample orientations (different from mirror planes), and displays strong non-sinusoidal oscillations of the center-of-mass curve.

The overall behavior can be understood by inspecting the Si band structure: HH5 (2.95 eV) is below the direct Si bandgap of 3.1 eV. This harmonic thus originates purely from intraband dynamics of low-energetic electrons, which mostly remain within

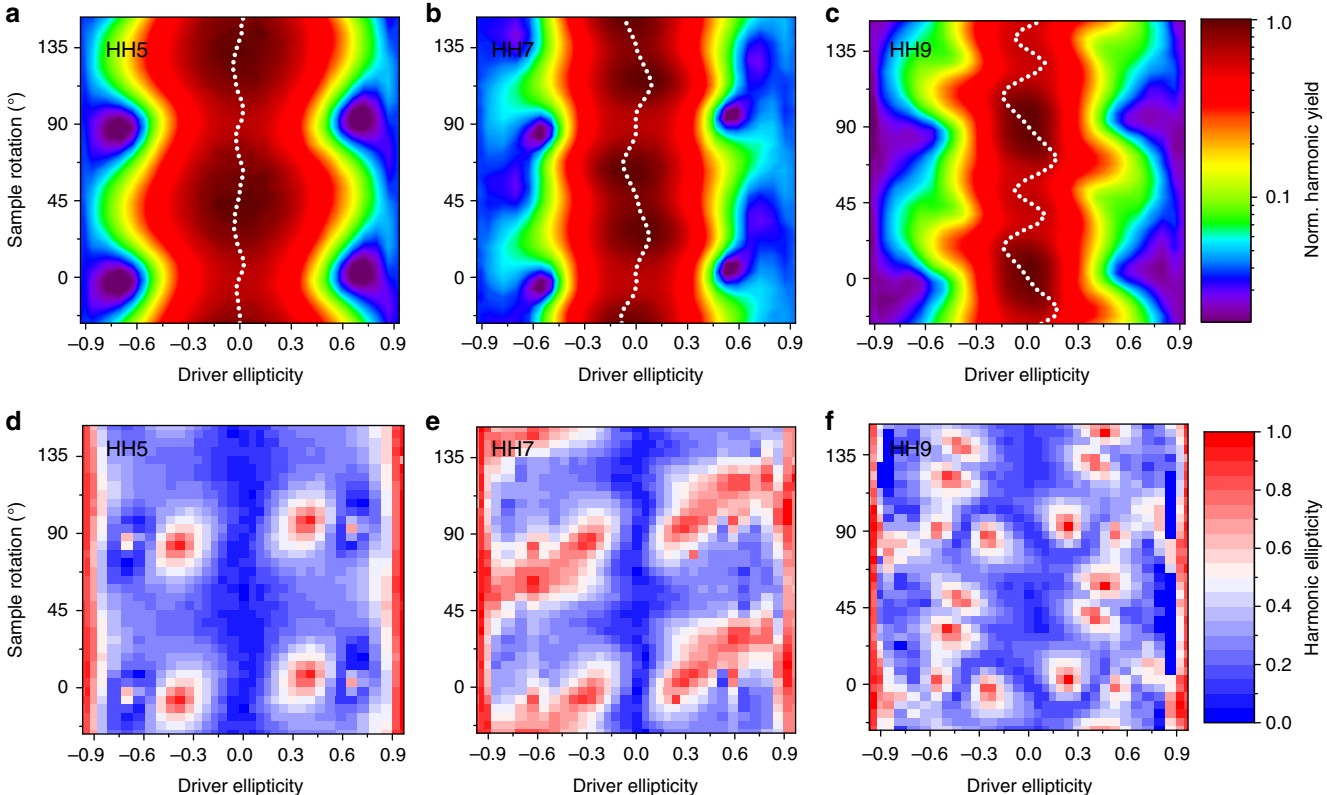

**Fig. 1** High-harmonic response of silicon versus driving pulse ellipticity and sample rotation. Measured intensity and harmonic ellipticity $|\varepsilon_{HH}|$ of HH5 (**a**, **d**), HH7 (**b**, **e**), and HH9 (**c**, **f**) as a function of driver ellipticity and sample rotation. The white dotted lines in **a**–**c** indicate the centers of mass (×5 to enhance visibility of the variation) of the intensity distributions. 0° and 90° sample rotation correspond to driver major axis along ΓX, 45° and 135° along ΓK. The peak driving intensity is 0.6 TW cm$^{-2}$ in vacuum

the parabolic region of the bands, leading to an atomic-like behavior. For above-bandgap harmonics, the joint density of states (JDOS) (see Supplementary Figure 6), i.e., the density of optical transitions at a given energy, determines the relative weight of interband compared to intraband mechanisms[14]. Around 5.3 eV (HH9), the JDOS is significantly lower than for 4.1 eV (HH7). Therefore, while coupled inter- and intraband dynamics lead to the emission of HH7, HH9 is mostly produced by intraband effects[14]. Interestingly, these harmonics are more efficiently generated with different helicities, as can be seen from the different signs of the center-of-mass curves for certain sample rotations (see Supplementary Figure 7). This clearly indicates different generation mechanisms of HH7 and HH9, as predicted in ref. [15]. For HH9, for which interband transitions are strongly suppressed by the low JDOS at this energy, higher-energetic electrons explore larger non-parabolic regions in the bands, which results in pronounced non-atomic-like ellipticity profiles.

Figure 1d–f reports the measured harmonics' polarization states as a function of driving ellipticity and sample rotation. Whereas linear drivers yield almost linear harmonics, we observe astonishing deviations of the harmonic ellipticities $\varepsilon_{HH}$ from the driver ellipticity $\varepsilon$. Consistent with our TDDFT predictions[15] and selection rules in ref. [28], for circular driver pulses, $|\varepsilon| \approx 1$, all harmonics become circular $|\varepsilon_{HH}| \approx 1$. Most importantly, for all observed harmonics, circular harmonics can be generated from elliptical driving polarizations, as elaborated on below. These islands of high ellipticity sensitively depend on $\varepsilon$ and $\theta$ in the cases of HH5 and HH9; however, for HH7 this sensitivity is less pronounced. This observation is again consistent with a strong dependence of the microscopic mechanisms on the polarization state of the driving

field, as the electrons explore different regions of the Brillouin zone (BZ) depending on $\varepsilon$ and $\theta$. The measured harmonics' polarizations contain the complete information on the x- and y-components of the harmonics' amplitudes and their relative phases.

Figure 2 summarizes our findings on circular harmonics from circular drivers. In both silicon (Fig. 2a) and α-quartz (Fig. 2b), all harmonic intensities remain constant while rotating a polarizer by 360°, thus confirming circular harmonic polarization. In Fig. 2c, we observe a strong intensity suppression of HH3 going from linear to circular driver, as expected from the selection rules for the $D_3[32]$ group[28] of α-quartz. The selection rules also manifest themselves in the helicities of the circular harmonics. In accordance with group-theoretical considerations[28] and TDDFT simulations[15], the odd harmonics from Si have alternating helicities as Si has point group $O_h$ [$m3m$] and is four-fold symmetric for our [001]-cut sample. This was confirmed with a tunable quarter-wave plate (QWP) behind the sample, which converts circular to linear polarization, with the polarization angle $\delta$ depending on the helicity (see Fig. 2d). The trigonal crystal structure of α-quartz results in different selection rules, leading to alternating helicities of HH4 and HH5 in Fig. 2e. As shown in Supplementary Note 9, we extracted the Stokes polarization parameters of the harmonics from these measurements and estimated a value of the degree of polarization of 0.8 ± 0.2 for all harmonics, similar to reported values for the generation of circular harmonics from atomic and molecular gases[29,30]. Moreover, we find in Si that the harmonic ellipticities $|\varepsilon_{HH}|$ are all close to 1, independent of sample rotation $\theta$ (see Fig. 2f). This isotropic behavior supports that driver pulses are almost perfectly circularly polarized.

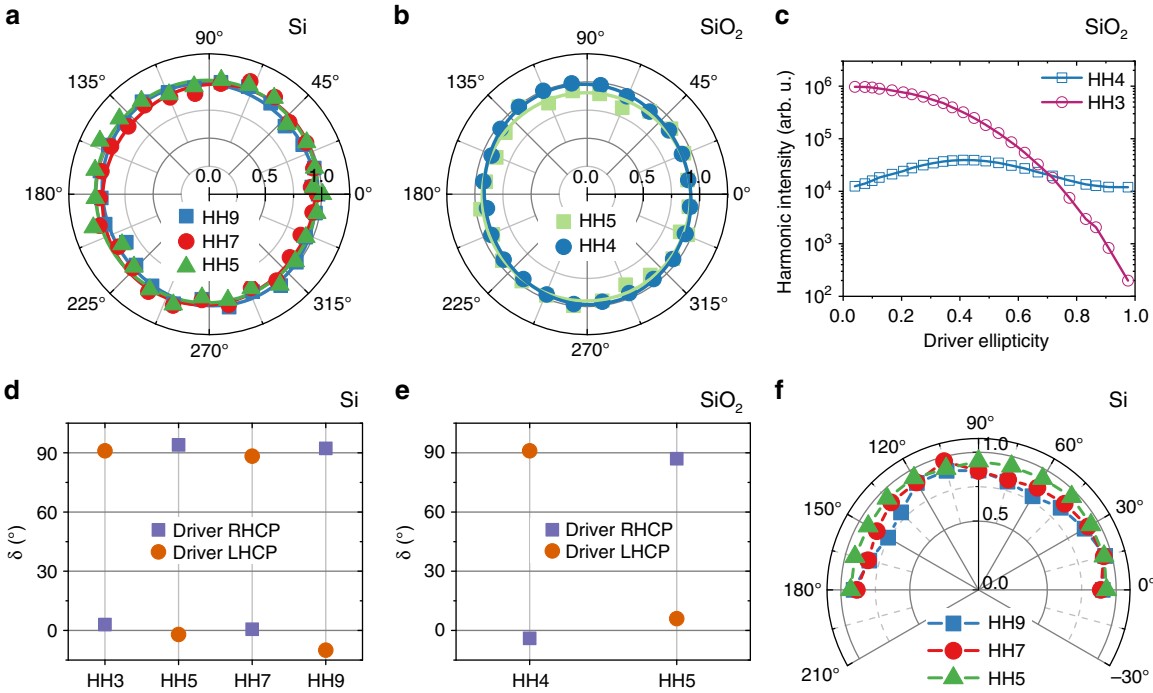

**Fig. 2** Measured circular harmonics from a circular driver and selection rules. Normalized harmonic intensity versus polarizer rotation angle from silicon (**a**) and quartz (**b**), showing circular harmonics. The solid lines are sin-square fits. **c** Intensity of HH3 and HH4 from quartz versus driver ellipticity. Harmonic major-axis rotation after a second quarter-wave plate for silicon (**d**) and quartz (**e**), indicating alternating helicities (RHCP/LHCP, right/left-handed circular polarization) with harmonic order, consistent with selection rules. **f** Harmonic ellipticities $|\varepsilon_{HH}|$ from Si versus sample rotation

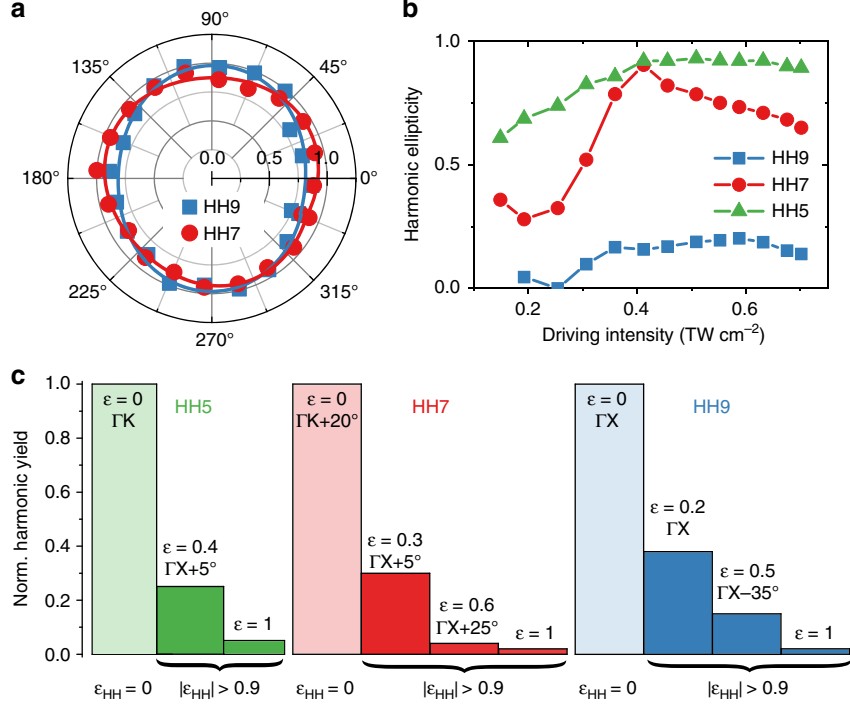

**Fig. 3** Measured circular harmonics from elliptical driver pulses in Si. **a** Polarizer scan of HH9 ($\varepsilon = 0.5$, $\theta = \Gamma X + 30°$) and HH7 ($\varepsilon = 0.3$, $\theta = \Gamma X + 5°$). The solid lines are sin-square fits. **b** Harmonic ellipticities $|\varepsilon_{HH}|$ versus driving intensity for $\varepsilon = 0.4$ and $\theta = \Gamma X + 10°$. **c** Yields of HH5–HH9 for exemplary cases of circular harmonic polarization. The harmonic yields are normalized to the maximum harmonic yield for $\varepsilon = 0$ (indicated by the light color bars)

Figure 3a shows two polarizer scans under excitation conditions, for which HH9 and HH7 are circular for elliptical driver polarization. The measured harmonic ellipticities $|\varepsilon_{HH}|$ are ~0.93 in both cases. We also found similarly high ellipticities for HH5 (see Supplementary Figure 8). Figure 3b shows the intensity dependence of the harmonic ellipticities for $\varepsilon = 0.4$ and $\theta = \Gamma X + 10°$. By varying the intensity of the driving field, we achieve a high degree of control over the harmonics' polarization

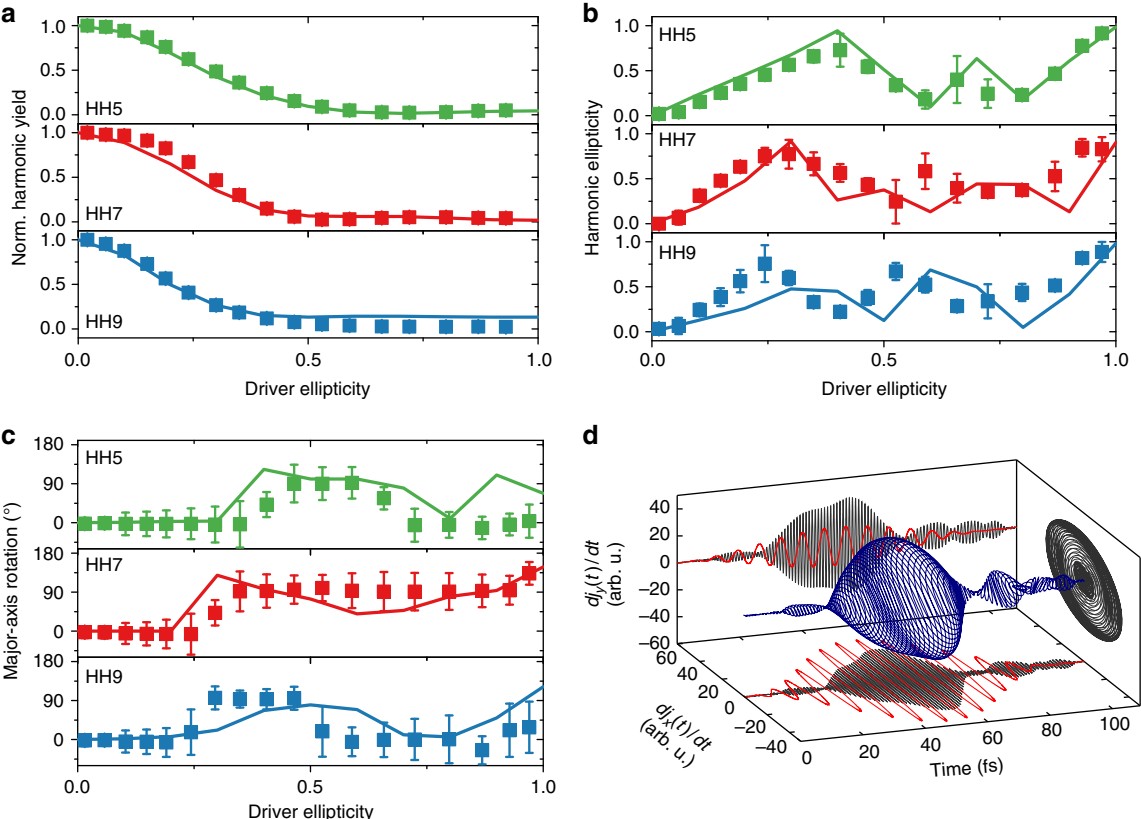

**Fig. 4** Calculated polarization states of the harmonics compared to the experiment. Comparison between TDDFT simulations (solid lines) and experimental results for **a** the harmonic yield, **b** the harmonic ellipticity $|\varepsilon_{HH}|$, and **c** the major-axis rotation of the harmonics' polarization ellipse of HH5 to HH9 versus the driver ellipticity. Here, $\theta = \Gamma X$. For all plots, the values are interpolated between $\theta = \Gamma X_{-3°}^{+2°}$ and averaged over negative and positive ellipticity values. The error bars are the averaged absolute deviations. **d** TDDFT result for the time-derivative of the electric current yielding HH7, for $\varepsilon = 0.3$ and $\theta = \Gamma X + 5°$. The red curves show the $x$- and $y$-projections of the driving laser field

states. This key result has two important consequences: First, it shows that the relative importance of interband and intraband mechanisms is not a material property only, but strongly depends on excitation conditions, thus offering a broader perspective on the controversial debate about the dominant mechanism responsible for HHG in solids. Second, the observation of circular harmonics for elliptical driver polarization, which sensitively depend on the nonperturbative dynamics of the system, can not be explained by symmetry arguments only, but clearly indicates strong-field control of the harmonic ellipticities $|\varepsilon_{HH}|$ through the lightwave-driven electron dynamics. This might find applications, e.g., in polarization-controlled high-harmonic sources.

The total harmonic intensities for exemplary cases of circular harmonics for different $\varepsilon$ and $\theta$ are compared in Fig. 3c. As discussed above, for Si, the harmonic yield tends to decrease (apart from non-monotonic exceptions) with increasing $|\varepsilon|$. Therefore, the generation of circular harmonics using elliptical driver pulses ($|\varepsilon| < 1$) is expected to be significantly more efficient than for circular ones ($|\varepsilon| = 1$), as indeed observed in Fig. 3c. For HH5 and HH7, the circular harmonics generated for $\varepsilon = 0.3–0.4$ and $\theta = \Gamma X + 5°$ are 10× brighter than for circular driver pulses. In the case of HH9, circular harmonics were even produced with 40% efficiency compared to maximum yield obtained for linear polarization, which corresponds to an 18× yield enhancement going from $\varepsilon = 1$ to $\varepsilon = 0.2$. We have also experimentally confirmed the harmonics' temporal and spatial coherence from Si (Supplementary Figs. 13 and 14), for both linear and circular harmonics. We found the

coherence time to be independent of the harmonics' polarization state. Circular harmonics exhibit spatial coherence.

Three scenarios are in principle possible to explain the observation of circular harmonics from elliptical driver pulses shown in Figs. 3 and 4: First, the harmonic emission occurs directly with this polarization state. Second, the harmonics are emitted with elliptical polarization and subsequently changed during propagation. Third, the driving pulse's polarization is altered during propagation due to induced birefringence. Moreover, the presence of the surface and a possible oxide layer might affect the polarization of the harmonics.

**Ab-initio TDDFT simulations**. To address this question, we performed extensive microscopic TDDFT simulations (see Methods section), which at this point do neither account for propagation nor surface effects, computing only the nonlinear microscopic response of the crystal to the incident electric field. For varying $\varepsilon$ and $\theta = \Gamma X$, we computed ab initio the high-harmonic response from Si and compared it to our measurements. The results shown in Fig. 4a–c display a remarkable agreement between experimental data and TDDFT calculations. This is true for harmonic yield, harmonic ellipticity as well as the rotation of the harmonics' major axes. We find minor deviations between calculations and experiments, mostly for HH7 and HH9, which can be expected by the increasing role of light propagation effects for photon energies above the bandgap. However, even in the presence of a surface and propagation effects in experiment,

the calculations yield circular harmonics from elliptical drivers exactly for the conditions in which they are observed experimentally. This is shown for HH7 in Fig. 4d. Therefore our ab-initio simulations confirm unambiguously that the measured polarization states of the harmonics have a microscopic origin in the coupled inter- and intraband dynamics, and is not due to macroscopic propagation effects or induced birefringence. From the comparison between experiments and simulations, it seems that the surface does not play a major role in determining the polarization states of the emitted harmonics.

## Discussion

In conclusion, after the first works on circular HHG from solids[15,21,22], we aimed at advancing our understanding and to demonstrate that a high degree of control over the polarization states of HHG from solids can be achieved. We found that both crystal symmetry and the nonperturbative coupled interband and intraband dynamics underlying harmonic emission play decisive roles in the polarization states of the emitted harmonics. We have elucidated this duality between symmetry and dynamics in experiments on high-harmonic generation from silicon and quartz accompanied by ab-initio TDDFT simulations. Our investigation has revealed that both the yields and polarization states of the higher harmonics sensitively respond differently to driver pulse ellipticity, sample rotation, and intensity. In a broader perspective, our results demonstrate that the relative importance of intraband and interband mechanisms is not only determined by the driving wavelength and the material itself, but can be dynamically controlled by the laser intensity.

Circular harmonics can be produced for both circular and elliptical driver polarizations: For circular driver pulses, the circular harmonics have alternating helicities, consistent with the selection rules derived from the crystallographic point-group symmetry[28]. For elliptical driver pulses, circular harmonics were generated for the first time to our knowledge, with up to 40% efficiency compared to linear driver pulses in Si, corresponding to an 18× enhancement compared to circular harmonics from circular drivers. Compact sources of bright circular harmonics from solids extending into the XUV regime might open up appealing new applications in the spectroscopy of chiral systems[18] and 2D materials with valley selectivity[11]. Circular isolated attosecond pulses from solids also seem in reach employing appropriate gating techniques. Finally, polarization-state-resolved high-harmonic spectroscopy offers the unique advantage of sensitivity to both electronic and structural dynamics on sub-cycle time scales, thus opening up new avenues for the spectroscopy of quantum materials on extreme time scales[3,23–26].

## Methods

**Experimental high-harmonic generation setup.** Supplementary Figure 1 shows the experimental setup used for HHG from crystalline solids. Passively carrier-envelope phase (CEP)-stabilized[31], 120-fs pulses at 2.1 μm (0.59 eV photon energy) are generated in a Ti:sapphire-pumped white-light-seeded optical parametric amplifier (OPA[32,33]. These 2.1-μm driver pulses pass through a wire-grid polarizer, a QWP and a half-wave plate, which allow setting the driver ellipticity while keeping the major axis of the polarization ellipse constant (see Supplementary Figure 2). The pulses are focused onto the sample with a 25-cm CaF$_2$ lens, resulting in a 1/$e^2$ focus diameter of 2$w_0 = 95$ μm. After 50 cm of propagation, an iris is used to spatially suppress the otherwise very strong third harmonic. A curved UV-enhanced Al mirror is used to direct the output light to an Ocean Optics UV-VIS HR4000 spectrometer with a slit width of 10 μm. To determine the ellipticities and major axes of the generated harmonics, a Rochon polarizer is placed between sample and iris and rotated in total by 360°, measuring a spectrum every 18°. To detect the helicity of the circular harmonics, a tunable zero-order QWP (from Alphalas) is placed between sample and polarizer. For post-processing the polarizer scans, the harmonic intensities are fitted with a sin-square curve offset from zero, the ellipticity calculated as $|\varepsilon_{HH}| = \sqrt{I_{min}/I_{max}}$ and the major-axis rotation as $\phi_{HH} = \arctan(I_y/I_x)$. The driving-intensity scan in Supplementary Figure 3 is performed employing reflective neutral-density filters.

**Ab-initio TDDFT simulations of high-harmonic generation in solids.** Within the framework of TDDFT, the evolution of the wavefunctions and the evaluation of the time-dependent current are computed by propagating the Kohn–Sham equations

$$i\hbar \frac{\partial \psi_{n,\mathbf{k}}(\mathbf{r},t)}{\partial t} = H_{KS}(\mathbf{r},t)\psi_{n,\mathbf{k}}(\mathbf{r},t), \quad (1)$$

where $\psi_{n,\mathbf{k}}$ is a Bloch state, $n$ a band index, $\mathbf{k}$ a point in the first Brillouin zone (BZ), and $H_{KS}$ is the Kohn–Sham Hamiltonian given by

$$H_{KS}(\mathbf{r},t) = \frac{1}{2m}\left(-i\hbar\nabla + \frac{e}{c}\mathbf{A}(t)\right)^2 + v_{ext}(\mathbf{r},t) + v_H(\mathbf{r},t) + v_{xc}(\mathbf{r},t). \quad (2)$$

The different terms correspond to the kinetic energy, the ionic potential, the Hartree potential, that describes the classical electron–electron interaction, and exchange-correlation potential, which contains all the correlations and nontrivial interactions between the electrons. The latter needs to be approximated in practice[34].

We perform the calculations using the Octopus code[35], employing the TB09[36] meta generalized gradient approximation (MGGA) functional to approximate the exchange-correlation potential using the adiabatic approximation. To ensure the stability of our time-propagation, we solved the time-dependent Kohn–Sham equations self-consistently at every time step using the enforced time-reversal symmetry propagator[37]. The c-value entering in the TB09 functional is recomputed at each time step using the gauge-invariant kinetic energy density. We employ norm-conserving pseudo-potentials. We emphasize that within TB09 MGGA, the experimental bandgap of common semiconductors and insulators is well reproduced[38], which is an important improvement over the local-density approximation (LDA) used in refs. [14,15], permitting direct comparison between experiment and theory. As shown in ref. [14] for adiabatic LDA (ALDA), local-field effects and dynamical correlations (at the level of the ALDA functional) do not seem to affect the HHG spectra of Si. The excitonic effects in Si mainly come from the long-range part of the exchange-correlation potential[39], i.e., a renormalization of the Hartree term (which does not play any role in HHG from Si[14]); therefore, excitonic effects are not expected to modify the HHG spectra of materials such as Si. We note that this is not necessarily true for all materials, in particular materials with strongly localized excitons, for which bound states will form in the bandgap, or in strongly correlated materials[23,24].

All calculations for bulk Si are performed using the primitive cell of bulk Si, using a real-space spacing of 0.484 atomic units, corresponding to 15 points along each primitive axis. We consider a laser pulse of 50-fs FWHM duration with a sin-square envelope and a carrier wavelength $\lambda$ of 2.08 μm, corresponding to 0.60 eV carrier photon energy. We employ an optimized 36 × 36 × 36 grid shifted four times to sample the BZ, and we use the intensity corresponding to the experimental intensity, using the value for the optical index $n$ of ~3.4 for computing the intensity in matter. The four shifts of the **k**-point grid are (in reduced coordinates) (0.5, 0.5, 0.5), (0.5, 0.0, 0.0), (0.0, 0.5, 0.0), (0.0, 0.0, 0.5). We use the experimental lattice constant $a$ leading to a MGGA bandgap (direct) of silicon of 3.09 eV. In all our calculations, we assume a CEP of $\phi = 0$.

We compute the total electronic current $\mathbf{j}(\mathbf{r}, t)$ from the time-evolved wavefunctions, the HHG spectrum is then directly given by

$$HHG(\omega) = \left| FT\left(\frac{\partial}{\partial t}\int d^3\mathbf{r}\, \mathbf{j}(\mathbf{r},t)\right)\right|^2, \quad (3)$$

where FT denotes the Fourier transform.

Supplementary Figure 5 shows a comparison of a computed HHG spectrum to a corresponding experimental spectrum. Note that, as mentioned above, in our experiments we only detect harmonics up to HH9 due to the spectrometer used (Ocean Optics UV-VIS HR4000). Our TDDFT calculations predict that harmonics up to HH19 in the XUV spectral region are generated for our experimental conditions.

## Code availability

The OCTOPUS code is available from http://www.octopus-code.org.

## Data availability

The data that support the findings of this study are available from the corresponding authors upon reasonable request, and will be deposited on the NoMaD repository (https://doi.org/10.17172/NOMAD/2019.03.04-1).

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

## Acknowledgements

We acknowledge support by the cluster of excellence "Advanced Imaging of Matter" (EXC 2056—project ID 390715994) and the priority program QUTIF (SPP1840 SOL-STICE) of the Deutsche Forschungsgemeinschaft, as well as financial support from the European Research Council (ERC-2015-AdG-694097), Grupos Consolidados (IT578-13), and European Union's H2020 program under GA no. 676580 (NOMAD). N.T.-D., A.R., and O.D.M. thank M. Altarelli for very fruitful discussion. We thank M. Spiwek for help with the Laue X-ray diffraction characterization of samples.

## Author contributions

N.T.-D., A.R., F.X.K., and O.D.M. conceived, designed, and coordinated the project. G.M.R. and R.E.M. implemented the IR-OPA driver source. N.K., Y.Y., G.D.S., and O.D.M. conceived the setup and performed the HHG experiments. F.S. and N.K. performed the 2DSI characterization. N.T.-D. carried out the code implementation and numerical calculations. N.K., N.T.-D., and O.D.M. analyzed and interpreted the experimental and theoretical results. N.K., N.T.-D., A.R., F.X.K., and O.D.M. participated in the discussion of the results and contributed to the manuscript with revisions by all.

## Additional information

**Competing interests:** The authors declare no competing interests.

