## [Peer Review File · Nature Communications]

REVIEWERS' COMMENTS:

Reviewer #1 (Remarks to the Author):

The authors responded to all the comments and questions by the reviewers carefully. The revised manuscript contains additional discussions and experimental results, which significantly improved its quality.

Observation and control of the polarization states of solid HHG is highly important for understanding its non-atomic behavior. The experimental and theoretical approach presented in the manuscript is versatile, thus can become a novel spectroscopy for various solids.

I therefore recommend the manuscript to be published in Nature Communications.

Reviewer #2 (Remarks to the Author):

Comments on the manuscript,

"Polarization-state-resolved high-harmonic spectroscopy of solids," by N. Klemke, N. Tancogne-Dejean, G. M. Rossi, Y. Yang, F. Scheiba, R. E. Mainz, G. Di Sciacca, A. Rubio, F. X. Kaertner, and O. D. Muecke.

In this manuscript, high-order harmonic (HH) generation from silicon and quartz is observed driven by TW/cm² class strong light field whose wavelength is 2.1 μm , well below the optical gap of the materials. The emitted harmonics is analyzed by means of polarization-state with sample rotation. Much stronger circularly polarized HHs are realized when the driver pulse satisfies conditions depending on the angle between the sample angle and ellipticity, compared to the circularly polarized driver. Ab-initio simulation based on time-dependent density functional theory (TDDFT) almost perfectly reproduces experimental signatures, harmonic yield, harmonic ellipticity, major-axis rotation depending on driving ellipticity, indicating the novel results due to solely microscopic response rather than light-propagation effects or the presence of the surface. The stronger circularly polarized HH with elliptic driver invokes fascinating applications as well as novelty itself. The ab-initio simulation gives us microscopic point of view to understand this phenomenon. This investigation demonstrates potential to control the polarization state of HH generation from solids.

I claimed requests and comments on the previous version of the manuscript to get ready for publishing. These are satisfactorily solved in the current manuscript. The revision according to other reviewer's comments seems quite reasonable for me. I would like to recommend the manuscript for publication in Nature Communications without further revision.

Reviewer #3 - only submitted comments to the Editor